# Enhanced Bioaccessibility and Antioxidant Activity of Curcumin from Transglutaminase Cross-Linked Mulberry Leaf Protein-Stabilized High-Internal-Phase Pickering Emulsion: In Vivo and In Vitro Studies

**DOI:** 10.3390/foods13233939

**Published:** 2024-12-06

**Authors:** Yingshan Xie, Hongyan Li, Zeyuan Deng, Yanfang Yu, Bing Zhang

**Affiliations:** 1State Key Laboratory of Food Science and Resource, Nanchang University, Nanchang 330047, China; 13543002174@163.com (Y.X.); lihongyan@ncu.edu.cn (H.L.); dengzy@ncu.edu.cn (Z.D.); 2International Institute of Food Innovation, Nanchang University, Nanchang 330051, China; 3Jiangxi Cash Crops Institute, Nanchang 330202, China; youyouyu325@163.com

**Keywords:** curcumin, TGase cross-linking, high-internal-phase Pickering emulsion, stability, bioaccessibility

## Abstract

The objective of this study was to formulate Pickering emulsions stabilized by transglutaminase cross-linked mulberry leaf protein (TG-MLP) nanoparticles as a delivery system for curcumin (Cur) and to assess its bioaccessibility both in vivo and in vitro. The encapsulation efficiency of curcumin in high-internal-phase Pickering emulsions (HIPEs) prepared at pH 10 with a 20 mg/mL concentration of TG-MLP reached 93%. Compared to Oil-Cur, Cur-HIPEs exhibited superior antioxidant activity. Furthermore, Cur-HIPEs demonstrated enhanced stability against ultraviolet irradiation, storage under dark and visible light, and heating, in contrast to Oil-Cur. Among the various conditions tested, HIPEs stabilized by TG-MLP nanoparticles at an ionic strength of 1000 mM offered the most effective protection for curcumin. Moreover, TG-MLP nanoparticles at pH 8 provided better stability for the formulated HIPEs compared to those at pH 6 and 10. During simulated gastrointestinal digestion, the bioaccessibility of curcumin in Cur-HIPEs was significantly increased to 30.1% compared to Oil-Cur. In murine studies, higher levels of curcumin were detected in the stomach, small intestine, rectum, ileum, and feces following administration of Cur-HIPEs, indicating improved protection, absorption, and potential biological activity during digestion. Consequently, HIPEs offer excellent protection and delivery for curcumin during digestion.

## 1. Introduction

High-internal-phase Pickering emulsions (HIPEs) are emulsions stabilized by solid particles irreversibly adsorbed onto an interfacial film. Compared to traditional emulsions stabilized by surfactants, HIPEs offer advantages such as high stability, low cost, and the ability to impart additional functionalities to the emulsion [1]. An HIPE is a super-concentrated emulsion where the dispersed phase exceeds 74%, forming polyhedral geometries due to droplet accumulation and deformation. This structure provides high drug loading and adjustable rheology, making HIPEs widely applicable in food, medicine, and cosmetics [1,2].

Pickering emulsions with high internal phases offer benefits such as resistance to caking and creaming, a lower emulsifier requirement (compared to the 5–50 wt% surfactant needed for traditional emulsions), and enhanced functionality when applied using templating methods [3]. Recently, HIPEs have gained attention in applications such as multi-margarine substitutes, functional material loading, and scaffold preparation. Phase inversion typically occurs when the oil-phase volume fraction of Pickering emulsion systems increases. Huang et al. utilized a chitosan–casein phosphopeptide (CS-CPP) nanocomplex as a particle emulsifier to stabilize HIPEs. The compact arrangement of CS-CPP at the oil–water interface inhibited oil oxidation. Additionally, curcumin’s bioaccessibility increased by 28.72% when loaded into this high-internal-phase Pickering emulsion compared to bulk oil [4]. Pickering emulsions stabilized by natural macromolecular organic polymers, such as proteins and polysaccharides, are non-toxic and safe and possess edible and nutritionally relevant properties [3].

The mulberry leaf is a promising protein resource due to its high protein content, comprising 17–25% protein (including all eight essential amino acids) on a dry matter basis [5]. Mulberry leaf protein, as a natural health product, holds significant development and research value. It demonstrates efficacy in reducing blood sugar and antioxidation, anticancer, anti-inflammatory, radical scavenging, and tyrosinase activity inhibition [6]. Research on protein particle-stabilized Pickering emulsion systems has primarily focused on animal proteins, soybean protein isolate, zein, sorghum protein, and other alcohol-soluble proteins. However, MLP, with its wide availability, high nutritional value, and strong functional properties, stands out as an excellent protein resource [7]. Notably, MLP is free from animal-based cholesterol, making it particularly popular among vegetarians and individuals needing to control cholesterol intake [8]. Additionally, MLP exhibits good gel-forming properties, which makes it suitable as a stabilizer for blowing agents and emulsifiers, broadening its potential applications in the food industry.

Enzymatic cross-linking is a crucial method for modifying proteins enzymatically. Unlike chemical modification, enzymatic cross-linking is favored in food processing due to its mild reaction conditions and high safety. Transglutaminase (TGase) is a commonly used enzyme for cross-linking food proteins. The acyl transfer reaction catalyzed by TGase facilitates the formation of ε-(γglutamyl)-lysine isopeptide covalent bonds between proteins and peptides, leading to cross-linked polymerization. This process enhances certain functional properties of proteins and holds promise in the food industry [9]. TGase has been shown to improve the hydration properties, emulsification properties, and thermal properties of vegetable proteins like soybean and peanut [9,10]. Babiker et al. reported that TGase-catalyzed cross-linked polymerization of soybean protease hydrolysates resulted in polymers with enhanced heat and acid resistance compared to directly cross-linked soybean protein. Additionally, their emulsification and foaming abilities were significantly improved [11]. Our laboratory previously found that a single mulberry leaf protein nanoparticle could not stabilize a high-internal-phase Pickering emulsion with an oil-phase volume fraction exceeding 74%, which necessitated modifying the properties of mulberry leaf protein. However, there is limited research on the TGase-mediated modification of mulberry leaf protein, which requires further investigation. Therefore, exploring TGase-induced cross-linking of mulberry leaf protein nanoparticles to stabilize HIPEs represents a novel and promising research avenue.

Curcumin, a polyphenol insoluble in water and slightly soluble in oil, has anti-inflammatory, antioxidant, and antibacterial properties, attracting attention across various fields [12]. However, its low solubility in water and susceptibility to rapid degradation and chemical instability under alkaline conditions, light, oxidation, and heat limit its industrial applications as a nutritional supplement or pharmaceutical ingredient [13]. In the pharmaceutical industry, improved bioaccessibility enhances curcumin absorption, enabling it to exert its anti-inflammatory, antioxidant, and other medicinal effects more effectively, which is valuable for developing drugs to treat inflammation-related or chronic diseases. Most stability studies on polyphenols have focused on microencapsulation. Microcapsules prepared from polysaccharides such as chitosan, sodium alginate, and maltodextrin can encapsulate polyphenols, enhancing their stability and bioaccessibility during production, processing, and utilization [14,15]. However, microcapsule preparation methods are often complex and susceptible to factors like temperature, leading to a loss of active substances [16]. HIPEs, due to their high internal phase ratio, can accommodate higher concentrations of hydrophobic active ingredients than ordinary oil-in-water emulsions. This provides a closed microenvironment for embedding lipophilic compounds like curcumin [17]. Lu et al. found that curcumin encapsulated in Pickering emulsions exhibited significantly higher bioaccessibility than bulk oil, demonstrating the potential of HIPEs as delivery systems for active ingredients in the food industry [18]. Additionally, natural vegetable oils rich in unsaturated fatty acids, such as corn oil with 85% unsaturated fatty acids, can enhance the antioxidant properties of Pickering emulsions, further improving the solubility and bioaccessibility of curcumin.

Therefore, this study aims to utilize TGase cross-linked mulberry leaf protein (MLP) nanoparticles to stabilize HIPEs as a novel encapsulation system for curcumin. Building on our previous study, which optimized the reaction conditions for TGase cross-linking of MLP, we selected pH 8.0, 50 °C, and 60 min for cross-linking to form TG-MLP polymers directly. We evaluated the ability of the Pickering emulsions to preserve curcumin under different pH and ionic strength conditions and investigated the antioxidant capacity of HIPEs. Furthermore, we explored the bioaccessibility of curcumin in Pickering emulsions during in vitro and in vivo digestion. To the best of our knowledge, this study represents the first attempt to encapsulate curcumin using TG-MLP-stabilized Pickering emulsions, shedding light on the mechanisms behind pH/ion-induced changes in curcumin retention and advancing the latest technologies in Pickering emulsion delivery carriers for curcumin. While curcumin served as a model lipophilic compound in this study, the insights gained could inform the rational design of nanogel-stabilized oil-in-water Pickering emulsions for encapsulating any lipophilic bioactive compound. This work offers new perspectives on bioactive substances and their applications in the functional food domain.

## 2. Materials and Methods

### 2.1. Material and Reagents

Mulberry leaves were provided by the Sericulture Tea Research Institute of Jiangxi Province. Purified water was obtained using the milliq Integral 3 system (Millipore, Darmstadt, Germany). Soybean oil was purchased from a local supermarket (LOTUS, Nanchang, China). Curcumin (purity ≥ 99%) was obtained from McLean Biochemical Technology Co., Ltd. (Shanghai, China). TGase (120 U/g), 1,1-diphenyl-2-trinitrophenylhydrazine (DPPH), and 2, 2-diazine-bis (3-ethyl-benzothiazole-6-sulfonic acid) diamiammonium salts (ABTS) were purchased from Aladdin Biochemical Technology Co., Ltd. (Shanghai, China). Pepsin, lipase, pancreatic enzyme, bile salt, sodium hydroxide (NaOH), hydrochloric acid (HCl), ammonium sulfate ((NH_4_)_2_SO_4_), methanol (chromatographic grade), acetic acid (chromatographic grade), anhydrous ethanol, potassium persulfate, and all other chemicals and solvents used in this study were of analytical grade and obtained from Xilong Science Co., Ltd. (Shantou, China).

### 2.2. Extraction of Mulberry Leaf Protein

The method for preparing MLP nanoparticles was adapted from previous protocols [7,19]. Mulberry leaf powder was dispersed in 0.3 mol/L NaOH at a solid–liquid ratio of 1:30 (*w*/*v*, g/mL) and continuously stirred at 75 °C for 1.5 h for protein extraction. The mixture was then centrifuged (Neofuge 15R, Lixin Instruments Co., Ltd., Shanghai, China) at 4000 rpm/min for 20 min to collect the supernatant. Ammonium sulfate was added to the supernatant to reach a concentration of 30% for protein precipitation, which was left overnight. The precipitated protein was then centrifuged for 20 min, dissolved, and reprecipitated with a small amount of distilled water. After 48 h of dialysis using a dialysis bag with a molecular weight cutoff of 8000–14,000 Da, the solution was centrifuged and filtered. A high-pressure microjet (M-110EH-30, Microfluidics International Corporation, Westwood, MA, USA) was used twice at a pressure of 120 MPa. The pH of the solution was adjusted to 3.8 for precipitation, followed by a 30-min incubation period before centrifugation. Finally, the sediment was freeze-dried using a vacuum freeze-dryer (ALPHA 1-2 LD Plus, Martin Christ Gefriertrocknungsanlagen GmbH, Martin Christ, Germany) to obtain MLP nanoparticles, which were stored in a dry environment.

### 2.3. Preparation of Mulberry Leaf Protein Cross-Linkers

The method for preparing mulberry leaf protein cross-linkers was adapted from previous protocols [20]. MLP nanoparticles were dispersed in distilled water to achieve a concentration of 20 mg/mL. The MLP nanoparticle suspension was adjusted to pH 12.0 and stirred at 28.07× *g* for 2 h. The mixture was then allowed to stand overnight to ensure adequate hydration. Afterward, the pH of the solution was adjusted to 8.0. Next, 20 U/g of TGase was added, and the reaction was carried out in a water bath at 50 °C for 60 min. The reaction was terminated by heating the mixture to 90 °C for 15 min, and the cross-linked MLP samples were then obtained through freeze-drying. A 20 mg/mL solution of the cross-linked MLP was prepared, labeled as TG-MLP.

### 2.4. Preparation of High-Internal-Phase Pickering Emulsion Loaded with Curcumin

The process for creating a high-internal-phase Pickering emulsion loaded with curcumin was based on previous protocols with modifications [21]. Curcumin was added to soybean oil to reach a final concentration of 1 mg/mL. The mixture underwent ultrasonication in the dark for 1 h, followed by magnetic stirring at 40 °C (800 rpm) overnight to ensure maximum dissolution of curcumin in the oil. After centrifugation at 12,000 rpm for 10 min to remove any undissolved solids, the supernatant was used as the oil phase for the Pickering emulsion, denoted as Oil-Cur. The TG-MLP dispersion solution prepared earlier was used as the aqueous phase, and it was homogenized using a high-speed homogenizer (IKA T18 Ultra-Turrax, IKA company, Staufen, Germany) at 16,000 rpm for 2 min. A high-internal-phase Pickering emulsion containing curcumin with an oil-to-water ratio of 2:8 was thus prepared, labeled as Cur-HIPEs. For comparison, a control Pickering emulsion was prepared using soybean oil without curcumin and the TG-MLP solution, denoted as TG-MLPPEs.

Sodium chloride was added to the TG-MLP solution (20 mg/mL, pH 8) to achieve ionic strengths of 300 mM, 500 mM, or 1000 mM. The high-internal-phase Pickering emulsion containing curcumin was then prepared by blending the TG-MLP dispersion solution with soybean oil at these different ionic strengths, followed by homogenization using a high-speed homogenizer (IKA T18 Ultra-Turrax, Staufen, Germany) at 16,000 rpm for 2 min. The pH of the TG-MLP solution was adjusted to 6, 8, and 10 using 0.1 mol/L solutions of NaOH and HCl. These TG-MLP dispersion solutions of varying pH levels were then mixed with soybean oil and homogenized to prepare the high-internal-phase Pickering emulsion containing curcumin.

### 2.5. Emulsion Droplet Size

The droplet size of the high-internal-phase Pickering emulsions prepared under various conditions was determined using a Mastersizer 3000 (Malvern Instruments, Malvin Instruments Ltd, Malvern, UK), as described in the method previously outlined in [22]. Milli-Q water (Merck Chemical Technology Co., Ltd, Shanghai, China) was used as the dispersant. The volume average droplet diameter (*D*_4,3_) of the emulsions was calculated as follows:(1)$$D_{4,3}=\sum n_{i}{d_{i}}^{4}/\sum n_{i}{d_{i}}^{3}\;$$

In the formula, *n_i_*—the number of droplets with diameter *d_i_*.

### 2.6. Optical Microscope Observation

After the emulsion was appropriately diluted, 50 μL of the emulsion was dispensed and placed on a slide. The slide was then covered, and the morphological characteristics of the emulsion were observed using an optical microscope under both 10× and 40× objective lenses.

### 2.7. Curcumin Loading Rate

The previous method was assessed and appropriately adjusted for extracting curcumin content from the sample using methanol [23]. To summarize, 1.8 mL of methanol was added to 200 μL of the emulsion, and the active ingredients were thoroughly extracted by vigorously shaking it with a vortex mixer for 2 min. The mixture was then centrifuged at 12,000 rpm for 10 min. The curcumin content in the supernatant was determined using a UV–visible spectrophotometer at 425 nm, with a methanol solution used as the blank. The curcumin content in the samples was calculated based on a standard curve. Each sample was measured in triplicate, and the results were averaged. The loading efficiency EE(%) of curcumin in the emulsion was calculated as follows:(2)$$\mathrm{EE}\left(\%\right)=\mathrm{Curcumin}\;\mathrm{content}\;\mathrm{in}\;\mathrm{emulsion}/\mathrm{Initial}\;\mathrm{Curcumin}\;\mathrm{content}\;\mathrm{added}\times 100\%$$

### 2.8. Rheological Properties

A rheometer (Anton Paar, MCR-102, Graz, Austria) was utilized to examine the rheological properties of the Pickering emulsion, following a previously established method with some modifications [24]. The fixture employed was a flat fixture (40 mm) with a 1 mm gap between the plates. The viscosity of the emulsion was measured as the shear rate increased from 0.1 to 100 s^−1^. Subsequently, the dynamic frequency of the emulsion was scanned from 0.1 to 10 Hz, with a fixed stress of 1 Pa, and the trend of the storage modulus (G′) and loss modulus (G″) with respect to frequency was recorded to characterize the viscoelasticity of the Pickering emulsion. All experiments were conducted at a test temperature of 25 ± 0.1 °C.

### 2.9. Antioxidant Properties

#### 2.9.1. Determination of DPPH• Free Radical Clearance

Referring to the methods described in the literature with some modifications [25], 0.045 g of Pickering emulsion was added to 1.8 mL of anhydrous ethanol, subjected to vortex mixing for 5 min to ensure complete dissolution, and then centrifuged at 14,000 rpm for 10 min. Subsequently, 500 μL of the supernatant was collected and mixed with 500 μL of DPPH working solution dissolved in anhydrous ethanol to achieve thorough mixing. The reaction mixture was left to stand in darkness at room temperature for 30 min. The supernatant was then mixed with anhydrous ethanol for further processing. Anhydrous ethanol mixed with the DPPH solution served as the blank control. The absorbance was measured at 517 nm. The percentage clearance of DPPH free radicals was calculated using the following formula:(3)$$\mathrm{DPPH}•\;\mathrm{free}\;\mathrm{radical}\;\mathrm{scavenging}\;\mathrm{rate}\left(\%\right)=\left(1-\left(A_{1}-A_{2}\right)/A_{0}\right)\times 100\%$$

In the formula, A_0_ is the absorbance of the DPPH• blank control; A_1_ is the absorbance of the sample reaction liquid; A_2_ is the absorbance of the sample without the DPPH• radical solution.

#### 2.9.2. ABTS• Determination of Free Radical Clearance

Referring to the methods described in the literature with some modifications [25], the solution was prepared as follows: a 7.4 mmol/L ABTS aqueous solution and a 2.45 mmol/L potassium persulfate aqueous solution were combined and left in the dark for 14 h to obtain an ABTS solution. Subsequently, the ABTS solution was diluted with a phosphate buffer solution (pH 5.8) until the absorbance of the solution at 734 nm reached 0.70 ± 0.072.

For the determination method, 0.045 g of the emulsion solution was mixed with 1.8 mL of anhydrous ethanol, subjected to vortex mixing for 5 min for complete dissolution, and then centrifuged at 14,000 rpm for 10 min. After centrifugation, 200 μL of the supernatant was collected and added to 1.4 mL of the prepared ABTS solution. The mixture was allowed to react at room temperature for 30 min in the absence of light, after which the absorbance was measured at 734 nm. A phosphate buffer with pH 7.4 was utilized as the blank control, and its absorbance value was recorded as A_0_. The clearance rate was used to indicate ABTS clearance ability, calculated using the following formula:(4)$$\mathrm{ABTS}•\;\mathrm{free}\;\mathrm{radical}\;\mathrm{scavenging}\;\mathrm{rate}\left(\%\right)=\left(1-\left(A_{1}-A_{2}\right)/A_{0}\right)\times 100\%$$

In the formula, A_0_ is the absorbance of the ABTS blank control; A_1_ is the absorbance of the sample reaction liquid; A_2_ is the absorbance of the sample without the ABTS radical solution.

### 2.10. Thermal Sterilization Stability

Pickering emulsion samples were subjected to heating at 30 °C, 60 °C, and 90 °C. For each temperature, 10 mL of emulsion was dispensed into test tubes and subsequently heated in a water bath. Each sample underwent heating for a duration of 5 h. Throughout the heating process, the samples were extracted every hour, and the content of active substances was determined according to the method outlined in Section 2.7.

### 2.11. Storage Stability

The emulsions containing curcumin were subjected to various ionic strengths and pH levels and stored under different environmental conditions. Specifically, they were stored at room temperature in darkness, exposed to visible light (λ = 400–700 nm), and subjected to ultraviolet radiation (6 W), while curcumin dissolved in soybean oil served as the control. The curcumin content of the emulsion samples under these diverse storage conditions was regularly assessed. Linear regression analysis was employed to estimate the time required for the curcumin content to decrease to 50% of its initial concentration, allowing for the calculation of the curcumin’s half-life. The degradation rate constant of curcumin in all samples was determined using a first-order kinetic model:lnC/C_0_ = −kt(5)
where C_0_ is the initial concentration of curcumin, C is the concentration of curcumin after ultraviolet radiation, t is the ultraviolet treatment time, and k is the degradation rate constant [26].

### 2.12. Analysis of Curcumin In Vitro Digestion

#### 2.12.1. Simulation of In Vitro Digestion Model

A model for the digestion of Cur-HIPEs in the in vitro gastrointestinal tract composed of oral, gastric, and intestinal phases was established according to the method described previously in [27].

Oral digestion: We used 2 mL of the samples diluted 4 times and preheated 2 mL of SSF at 37 °C for 5 min before mixing. The pH of the mixture was quickly adjusted to 6.8 with 1 mol/L NaOH and digested in a constant-temperature shaking bath (100 r/min) at 37 °C for 10 min.

Gastric digestion: After preheating SGF at 37 °C for 5 min, we mixed 3 mL of SGF with the previously orally digested samples. Then, we added 0.5 mL each of pepsin and lipase. The pH of the mixed system was quickly adjusted to 2.5 with 1 M HCl and digested at 37 °C in a constant-temperature shaking bath (100 r/min) for 2 h.

Intestinal digestion: Following gastric digestion, we quickly adjusted the pH of the mixed system to 7.0 using 2 mol/L NaOH. After preheating SGF at 37 °C for 5 min, we mixed 7 mL SIF with the previously gastric digested samples. Subsequently, we added 0.5 mL of pancreatic enzyme and 0.5 mL of bile salt sequentially and adjusted the pH of the mixed system to 7.0 with 1 mol/L NaOH. Finally, it was digested in a water bath (100 rpm) at 37 °C for 2 h. We then placed it in a 90 °C water bath for 5 min to inactivate the pancreatic enzymes and terminate digestion.

#### 2.12.2. Determination of Bioaccessibility of Curcumin

Following the method outlined by Wei et al. [21], the simulated in vitro digestion was conducted. After digestion, the digestive fluid was collected and centrifuged at 4 °C for 30 min to separate it into three layers. The upper layer consisted of undigested oil, the lower layer contained hard-to-digest or undigested solids, and the transparent portion of the middle layer represented the micellar layer. To isolate curcumin, 4 mL of the micellar phase was transferred into a centrifuge tube, followed by the addition of 4 mL of methanol solution. The mixture was then swirled and shaken for 4 min to ensure uniformity, after which it was centrifuged at 14,000 rpm for 10 min. This process was repeated until curcumin was completely released from the micellar phase. The content of curcumin in the micellar layer was determined using high-performance liquid chromatography (HPLC), and the bioaccessibility (%) of curcumin was calculated using the following formula:Bioaccessibility (%) = curcumin content in micelles/initial curcumin content in samples × 100%(6)

The determination of curcumin by HPLC followed the method described by Wei et al. [21]. The mobile phase consisted of methanol/0.2% acetic acid water in a ratio of 95:5, with a flow rate of 1 mL/min. Samples of 20 μL were injected and analyzed using an Agilent XDS-C18 column (250 × 4.6 mm, i.d. 5 mm), with the peak area of curcumin recorded at 425 nm. The content of curcumin in curcumin samples with different concentrations (ranging from 1.0 to 50.0 μg/mL) was determined using an HPLC standard curve constructed under the same conditions.

Preparation of the curcumin standard solution involved weighing 10 mg of the curcumin standard and dissolving it in a methanol solution to prepare a 1 mg/mL curcumin standard reserve solution. This solution was then diluted with methanol to prepare a series of standard solutions with concentrations of 1.0, 2.0, 5.0, 10.0, 20.0, and 50.0 μg/mL. The peak area under different concentrations was determined by HPLC, and a standard curve was plotted with the peak area as the vertical coordinate (y) and curcumin concentration as the horizontal coordinate (x).

### 2.13. Release of Cur-HIPE in Mice

#### 2.13.1. Animal Feeding

Male C57BL/6J mice (6-week, 18–24 g) were purchased from Nanjing crisbio Biotechnology Co., Ltd (Nanjing, China) (SCXK-JIN (019-0010). The mice were housed in plastic cages under conditions of 50–55% humidity, a temperature of 22 ± 2 °C, and a 12-h light/dark cycle.

#### 2.13.2. Experimental Process

Mice were grouped following adjustments based on previous studies [27]. Mice were divided into 4 groups including blank, control, Oil-Cur, and Cur-HIPEs. These groups received normal saline, HIPEs without curcumin, Oil-Cur (soybean oil dissolved with curcumin), and Cur-HIPEs via gavage, respectively. After a one-week adaptation period with a standard diet, the mice were adaptively fed with common chow for an additional week before gavaging. Subsequently, mice were gavaged according to their respective groups, with each mouse receiving 0.2 mL per gavage. The blank and control groups were euthanized 4 h after administration. Mice in the Oil-Cur and Cur-HIPEs groups were intragastrically administered and euthanized at 0.5, 1, 2, and 4-h time points. Feces were collected, and gastrointestinal tracts were dissected to obtain stomach, small intestine, ileum, rectum, and fecal samples. All animal experiments were conducted in strict accordance with China’s Experimental Animal Welfare Ethical Review Guidelines (GB/T 35892-2018 [28]) and approved by the Ethics Committee of Nanchang University (Permit Number: 0064257).

#### 2.13.3. Determination of Curcumin Content by Mass Spectrometry

Appropriate amounts of mouse stomach contents, small intestine, ileum and rectal tissue contents, and feces were weighed and recorded. A total of 1 mL of methanol was added, followed by grinding in a tissue grinder until no solid was visible. The mixture was then centrifuged at 12,000× *g* for 15 min, and the supernatant was collected. After nitrogen blowing or vacuum centrifugation, the liquid was concentrated, dried, redissolved in an appropriate amount of methanol, and passed through a 0.22 μm organic filter membrane.

Quantitative analysis was performed using the Q Exactive Focus mass spectrometer (Thermo Fisher, Waltham, MA, USA) and the Vanquish HPLC system (Thermo Fisher, Waltham, MA, USA) in positive ion mode using PRM. The parent and daughter ions selected were 369.13 and 177.05, respectively. The injection volume was 2 μL. The mobile phases A and B consisted of 0.2% acetic acid water and methanol solution, respectively. A chromatographic column was used (100 × 2.1 mm, 2.6 μm; Thermo Fisher Scientific, Bremen, Germany), employing a methanol/0.2% acetic acid water = 95:5 gradient to achieve chromatographic separation at a flow rate of 0.2 mL/min. The mass spectrum conditions were set with a capillary temperature and auxiliary gas heater temperature of 300 °C and 330 °C, respectively, and a spray voltage adjusted to 3 kV. The acquisition quality range was set to 50–500 Da, and the S lens RF level was set at 25.

Qualitative analysis was conducted using the Q Exactive Focus mass spectrometer (Thermo Fisher, Waltham, MA, USA) and the Vanquish HPLC system (Thermo Fisher, Waltham, MA, USA) in Full MS mode for positive ion samples, with Compound Discover used for data analysis.

### 2.14. Statistical Analysis

Each sample was tested in triplicate, and the data were processed as the mean ± standard deviation. Statistical analysis was performed using Origin Pro 9.0.5 software, with differences between groups assessed by one-way analysis of variance (ANOVA) with a significance level set at *p* < 0.05.

## 3. Results and Discussion

### 3.1. Emulsion Droplet Size

Ionic species not only diminish the surface potential of colloidal particles but also promote their condensation into flocculants. This investigation assesses how varying concentrations of NaCl (0–1000 mM) influence the stability of high-internal-phase emulsions (HIPEs). This study reveals that the emulsion droplet sizes of both curcumin-loaded HIPEs (Cur-HIPEs) and traditional gel–MLP polyethylene emulsions (TG-MLPPEs) remain comparable in the absence of a salt solution. Generally, the addition of curcumin does not modify the system’s droplet size at the micron scale [29]. This effect is notable only in nanoemulsions, where the size and concentration of curcumin crystals significantly influence droplet size changes. Previous studies confirm that incorporating curcumin into surfactant-stabilized nanoemulsions increases their average droplet size, thereby destabilizing the emulsion [30]. The negligible difference in HIPEs’ droplet size before and after curcumin incorporation suggests the relative insignificance of curcumin crystal size to the emulsion. Consequently, curcumin-coated HIPEs’ droplet size remains largely unaffected [13], indicating the nanoparticles’ efficient encapsulation of active substances without disrupting the emulsion’s gel network structure.

Figure 1A demonstrates that increased ionic strength leads to larger emulsion droplets, suggesting that salt addition mitigates charge repulsion among protein particles, facilitating denser protein assemblies at the oil–water interface and thereby explaining flocculation phenomena [31]. Despite flocculation, no precipitation is observed even at high salt concentrations, attributed to the droplets’ significant charge inducing strong electrostatic repulsion. This finding is corroborated by optical microscopy (Figure 1C) and aligns with Shah et al. [23], indicating that droplet size varies slightly with salt addition, likely due to an enhanced electrostatic shielding effect.

The impact of pH on Pickering emulsion stability, critical during food processing and digestion, was also examined [32]. pH significantly influences the wettability and charge of polyelectrolyte particles, thereby affecting emulsion stability. As depicted in Figure 1B, emulsion droplets are smaller at pH 8 and pH 10, with a notable size increase at pH 6 due to larger particle agglomerations at this lower pH, attributed to decreased electrostatic repulsion [33]. This observation is supported by optical microscopy (Figure 1C). Figure 1D presents the appearance of curcumin-loaded high-internal-phase Pickering emulsions under various conditions after 30 days, showcasing stable, uniformly bright yellow emulsions without oil leakage or emulsion breaking, signifying excellent storage stability.

### 3.2. Rheological Characteristics

Regarding rheological properties, as illustrated in Figure 2A,B, both TG-MLPPEs and curcumin-loaded Pickering emulsions exhibit shear-thinning behavior, indicating that curcumin embedding does not compromise the non-Newtonian fluid characteristics of Pickering emulsions. Curcumin incorporation notably enhances the emulsions’ viscoelasticity, suggesting improved stability. This finding is consistent with Xia et al. [34], who observed similar enhancements in viscoelasticity upon embedding curcumin using lactoferrin as a carrier in high-internal-phase Pickering emulsions.

As illustrated in Figure 2C,D, under varying ionic strengths and pH conditions, the TG-MLP nanoparticle-stabilized high-internal-phase Pickering emulsions display gel properties, with the storage modulus (G′) consistently surpassing the loss modulus (G′′), indicating the formation of solid-like elastic gel structures [35]. This gelation is influenced by pH, with the initial G′ values being highest at pH 8, followed by pH 6, and lowest at pH 10, suggesting pH-dependent gel strength and stability [36]. These rheological properties are crucial for understanding the emulsions’ ability to retain curcumin under different conditions.

### 3.3. Analysis of Antioxidant Capacity

In the food industry, curcumin’s strong antioxidant properties can help delay food oxidation and prevent the oxidative rancidity that commonly occurs in oil-rich foods, thereby extending their shelf life. Additionally, as a natural ingredient, curcumin aligns with consumer demand for healthy, natural additives. In this study, curcumin was dissolved in soybean oil to serve as the dispersing phase for preparing high-internal-phase Pickering emulsions. The successful incorporation of curcumin into the emulsion system imparted antioxidant properties to the emulsion. To assess the protective effect of TG-MLP encapsulation, the antioxidant activity of curcumin was characterized by its DPPH and ABTS free radical scavenging activities before and after 30 days of storage. High-internal-phase Pickering emulsions containing curcumin, stabilized using TG-MLP at pH 8, were designated as Cur-HIPEs. For comparative analysis, high-internal-phase Pickering emulsions without curcumin (TG-MLPPEs) and curcumin dissolved in soybean oil (Oil-Cur) were used as control groups to evaluate the antioxidant capacity.

As depicted in Figure 3, TG-MLPPEs exhibited DPPH and ABTS free radical scavenging rates of 85.7% and 87.5%, respectively, showcasing potent antioxidant activity. This can be attributed to phenylalanine, threonine, tyrosine, and serine in mulberry leaf protein possessing numerous active phenolic hydroxyl groups conjugated with double bonds, thereby exhibiting robust antioxidant activity [37]. With prolonged storage, the DPPH and ABTS free radical scavenging rates decreased. After 30 days, the DPPH and ABTS free radical scavenging rates of Cur-HIPEs decreased from 93.9% to 81.7% and from 94.6% to 83.3%, respectively, representing a 13.0% and 12.0% decrease, respectively. Despite this decrease, the scavenging rates remained substantial. Additionally, the emulsions’ thick interface acted as an oxygen barrier, and storage in dark conditions mitigated curcumin degradation, thus preserving its antioxidant activity.

In contrast, Oil-Cur, containing free curcumin, exhibited higher initial DPPH and ABTS free radical scavenging rates. However, after 30 days of storage, these rates decreased significantly by 27.4% and 28.9%, respectively, indicating rapid curcumin degradation and poor stability over time. Studies suggest that the larger particle size and thicker interface of high-internal-phase emulsions afford better protection against oxidation compared to pure oil-based curcumin systems [38].

### 3.4. Thermal Stability

As illustrated in Figure 4, the degradation rate of curcumin was notably slower in Cur-HIPEs compared to Oil-Cur, highlighting the protective effect of the emulsion on curcumin. It was observed that curcumin content decreased more severely after heating at 90 °C than at 30 °C and 60 °C. At 90 °C, the emulsion provided no significant protection for curcumin over time. This observation is consistent with the Arrhenius rate law, which states that reaction rates increase exponentially with temperature [39]. At higher temperatures, molecular collisions occur more frequently, leading to accelerated degradation of curcumin [40]. Young et al. reported that only 66% of curcumin was recovered from a silica-stabilized emulsion stored at 70 °C for 30 min [41]. This result suggests that high-temperature storage presents the greatest risk for curcumin degradation under various conditions. Therefore, to optimize curcumin stability, it is recommended to store emulsions at lower temperatures.

### 3.5. Storage Stability

#### 3.5.1. Protective Effect of Stable Pickering Emulsion on Curcumin Under Different Ionic Strengths

Given the potential of Pickering emulsions as effective carriers for nutraceutical delivery and the protective capacity of high-internal-phase emulsions (HIPEs) for sensitive nutraceuticals [42], the chemical stability of Cur-HIPEs was explored to assess the shielding effect of TGase cross-linked mulberry leaf protein-stabilized HIPEs. Considering the variation in ionic strength in emulsified foods, investigating the protective influence of TGase cross-linked mulberry leaf protein-stabilized HIPEs on curcumin under different ionic strengths is crucial.

Appendix A illustrates the loading rate of HIPEs for curcumin under varying ionic strength and pH conditions. As depicted in Appendix A, in the absence of added salt ions, the loading rate of HIPEs for curcumin is optimal, reaching 91%. This phenomenon arises due to the electrostatic shielding effect induced by increased ionic strength, which results from the ionic cross-linking between salt ions and protein particles, subsequently reducing the Zeta potential. Consistent with prior research, the augmentation in ionic strength leads to a gradual increase in particle size and a reduction in the Zeta potential of soybean proteins [43]. Consequently, the emulsion’s stability diminishes, leading to the precipitation of curcumin from the emulsion and a decrease in the loading rate. In Appendix A, it is evident that the loading rate of curcumin steadily rises with increasing pH. Previous studies have demonstrated that protein solubility increases with rising pH values [7]. At a pH of 10, the loading rate can reach 93%. This is attributed to the favorable solubility of TG-MLP in alkaline environments, preventing particle aggregation and maintaining stability, thereby facilitating the preparation of HIPEs with excellent stability and loading rates.

Ultraviolet (UV) irradiation is a common method for sterilization in the food and drug industry. However, curcumin, being a polyphenol, is vulnerable to UV light-induced degradation. Under light exposure, curcumin absorbs energy from UV light, leading to molecular alterations characterized by the disruption of adjacent hydroxyl groups, resulting in the formation of organic aldehydes, organic acids, and vanillic acids [44,45]. Therefore, the degree of curcumin retention post-UV radiation serves as an indicator of the protective effect of TG-MLP-stabilized HIPEs.

Table 1 summarizes the degradation rate constant and half-life of curcumin of HIPEs under ultraviolet, dark, and visible light at different ionic strengths. The degradation rate constant of curcumin follows this order: emulsion with an ionic strength of 1000 mM < emulsion with an ionic strength of 500 mM < emulsion with an ionic strength of 300 mM < emulsion with an ionic strength of 0 mM < soybean oil. Over time, curcumin content in the emulsion gradually decreases, following the aforementioned trend (Figure 5A).

Curcumin in pure oil exhibits the fastest degradation rate during UV irradiation due to the unrestricted exposure of curcumin molecules to UV light, facilitating efficient energy absorption for degradation. Conversely, in the emulsion system, the TG-MLP particle system envelops the oil droplets, forming a UV radiation barrier, limiting degradation reactions to the interface. Consequently, the protective effect of the interface layer shields curcumin from degradation.

Increasing ionic strength in TG-MLP-stabilized Pickering emulsions enhances curcumin protection. This can be attributed to salt addition weakening the charge repulsion effect between TG-MLP nanoparticles, ensuring sufficient TG-MLP coverage at the oil–water interface, thereby forming a thicker and denser TG-MLP layer around oil droplets and enhancing curcumin protection [31]. Thus, the TG-MLP-stabilized Pickering emulsion at 1000 mM ion strength exhibits superior curcumin protection across all studied ion strengths.

This study also investigated the effects of storing Pickering emulsions in darkness and under visible light, with varying ionic strengths, on curcumin retention. The degradation rate constant of curcumin was calculated using a first-order kinetic model, and the results were consistent with those observed under UV irradiation. Notably, curcumin degradation in the dark was slower than under visible light, likely due to curcumin’s inherent instability when exposed to light. Storage in both dark and visible light conditions resulted in decreased curcumin retention in all samples (Figure 5C,D). This highlights the protective role of the MLP layer surrounding the oil droplets in reducing physicochemical degradation and preserving curcumin retention. Similar findings have been reported in previous studies, which demonstrated that UV-induced curcumin degradation in HIPEs can be mitigated by using zein/pectin coatings around oil droplets [46].

#### 3.5.2. Protective Effect of TG-MLP-Stabilized Pickering Emulsion on Curcumin at Different pH Levels

This study examined the impact of TG-MLP-stabilized Pickering emulsions stored under UV, dark, and visible light conditions at varying pH levels on curcumin retention. A first-order kinetic model was employed to determine the degradation rate constant and half-life of curcumin across all samples (Table 2). As illustrated in Figure 5E, curcumin within TG-MLP-stabilized high-interior-phase Pickering emulsions exhibited slower degradation rates compared to curcumin dissolved in soybean oil across different pH values, indicating that the interface layer composed of TG-MLP serves as a protective barrier against UV radiation. The residual curcumin levels in TG-MLP-stabilized Pickering emulsions post-UV treatment followed the order pH 8 emulsion > pH 6 emulsion > pH 10 emulsion, indicating that emulsions at pH 8 offer the most effective protection for curcumin.

The size of emulsion droplets can impact the volume-to-surface area ratio of the emulsion. With a fixed total volume of emulsion droplets, reducing their size may increase the total surface area of the emulsion droplets [31]. Curcumin in emulsions with larger interfacial areas may degrade more rapidly. As demonstrated, the TG-MLP-stabilized emulsion at pH 10 exhibited a smaller droplet size compared to that at pH 6. This elucidates why lotions with a pH of 10 offer inferior protection. It is worth noting that while TG-MLP-stabilized emulsions at pH 8 possess the smallest droplet size and the largest interfacial area among those at different pH values, the beneficial effects on curcumin protection attributable to the denser interfacial protein layer might outweigh the adverse effects of the larger interfacial area. Some scholars have also arrived at similar conclusions [31]. Based on these findings, TG-MLP-stabilized emulsions at pH 8 present the most effective interface barrier for curcumin protection, showcasing promising potential as suitable delivery vectors for curcumin.

As shown in Figure 5F,G, curcumin retention decreased in all samples under dark and visible light storage conditions, indicating that the MLP layer surrounding the oil droplets may help mitigate physicochemical degradation and maintain high curcumin retention. The degradation rate of curcumin was slower in the dark compared to visible light conditions (Table 2), which is likely due to the instability of curcumin when exposed to light. This observation is consistent with findings from other studies, which have also reported that light exposure accelerates curcumin degradation [47].

### 3.6. Bioaccessibility In Vitro

Curcumin, as a bioactive substance, holds significant potential in functional food applications. Enhancing its stability within the digestive system is crucial for maximizing its utility. Thus, the protective efficacy of HIPEs on curcumin was assessed through in vitro simulation of the digestive system. A Pickering emulsion prepared at pH 8 with an ionic strength of 0 mM served as the experimental group, while curcumin dissolved in soybean oil acted as the control group. The chemical instability of curcumin significantly impacts its bioactivity, potentially affecting its bioaccessibility within the gastrointestinal tract [48]. Following simulated in vitro digestion, lipids undergo hydrolysis into diglycerides, free fatty acids, and other components, which subsequently form micelles with bile salts, phospholipids, etc. Curcumin initially dissolves in fats and then dissolves into these micelles, enhancing its bioaccessibility. High-performance liquid chromatography (HPLC) was utilized to assess the bioaccessibility of curcumin dissolved in micelles post-digestion. Bioaccessibility hinges on both the amount of the original material dissolved in the micelle and the amount of post-chemical conversion.

As observed in Figure 6A, the bioaccessibility of curcumin within the Pickering emulsion is significantly higher than that of curcumin directly dissolved in oil. The bioaccessibility of curcumin in oils alone is low, at only 15.5%. Upon loading into the emulsion, the bioaccessibility of curcumin increases to 30.1% and 94.2%, respectively, indicating that the emulsion system effectively enhances curcumin bioaccessibility to a considerable extent. This enhancement can be attributed to the Pickering emulsion’s elastic gel structure, which retards curcumin degradation during in vitro digestion, reduces oil droplet aggregation, and augments the contact area and reaction rates between lipase and oil droplets, thereby enhancing lipolysis. Greater lipolysis yields more micelles, facilitating increased curcumin release into the micellar phase [49]. In contrast, in oils and fats, with their large droplets and limited specific surface area, the interaction between lipase and oil droplets is insufficient, thereby impeding the transformation of curcumin from oils to micelles [50,51].

### 3.7. Bioaccessibility In Vivo

Based on the findings from the in vitro digestion experiments, further investigation into the effects of high-internal-phase emulsions (HIPEs) on the protection, delivery, and resistance to digestion of curcumin in vivo was conducted. Curcumin content in the stomach contents, small intestine tissues, ileum, rectal contents, and feces of mice was measured. It is worth noting that curcumin was undetectable in both the blank group and the control group, confirming that interference from other substances in mice did not affect the detection of curcumin.

As shown in Figure 6B, curcumin content in the stomach contents of mice gradually decreased over 4 h, with the Cur-HIPE group exhibiting higher levels compared to the Oil-Cur group at each time point. Previous studies have indicated that curcumin is susceptible to degradation in acidic environments [52], indicating that HIPEs offer protection against stomach acidity. In small intestine tissues (Figure 6C), curcumin content decreased over 4 h in the Oil-Cur group but showed an initial increase followed by a decrease in the Cur-HIPE group. This suggests a rapid increase in oil content in the small intestine, peaking after 0.5 h, while the emulsion remains present for a longer duration, peaking at 2 h. In the ileum tissue (Figure 6D), curcumin content in the Oil-Cur group initially increased and then decreased over 4 h, whereas in the Cur-HIPE group, it exhibited a continuous increase, indicating prolonged presence of oil in the ileum without reaching a peak.

As shown in Figure 6E, both the Oil-Cur and Cur-HIPE groups exhibited increasing curcumin content in rectal tissue over the 4-h period, suggesting the continued presence of oil and HIPEs in the rectum. Figure 6F shows that curcumin content in feces also increased over the same period for both groups, indicating the ongoing excretion of oil and HIPEs. In summary, compared to oil, HIPEs remained in the mice for a longer duration, thereby extending the presence of curcumin in the body. This observation aligns with findings from the existing literature which have also reported prolonged retention of curcumin in the body when delivered via emulsions [53].

## 4. Conclusions

In summary, the encapsulation efficiency of curcumin in high-internal-phase Pickering emulsions (HIPEs), prepared at pH 8 with a 20 mg/mL TG-MLP concentration, reached 89.88%. Various TG-MLP-stabilized emulsions with different ionic strengths and pH levels were formulated and analyzed for droplet size, microstructure, rheological properties, and stability. Compared to Oil-Cur, Cur-HIPEs exhibited enhanced antioxidant capacity and significantly improved stability under ultraviolet irradiation, dark and visible light storage, and heating conditions. When incorporated into HIPEs, curcumin demonstrated enhanced bioaccessibility during in vitro gastrointestinal digestion simulation. Furthermore, curcumin in HIPEs was well protected during gastric digestion, released slowly during intestinal digestion, and showed increased absorption in the small intestine during digestion in mice. Therefore, food-grade HIPEs offer a promising approach for protecting curcumin during storage while simultaneously enhancing its nutritional and biological properties. The use of natural, plant-based proteins like mulberry leaf protein (MLP) presents an eco-friendly and sustainable alternative to synthetic emulsifiers, which are commonly used in the food and pharmaceutical industries. As consumer demand shifts toward plant-based and sustainable ingredients, this technology is well suited to meet market trends for clean-label products and greater environmental awareness. Future research should focus on developing cost-effective production methods, utilizing large-scale manufacturing processes, and assessing consumer acceptance to drive the commercialization of curcumin-based emulsion systems. The successful commercialization of this technology could lead to the creation of more efficient, eco-friendly, and health-promoting products that cater to the increasing consumer demand for natural and functional ingredients.

## Figures and Tables

**Figure 1 foods-13-03939-f001:**
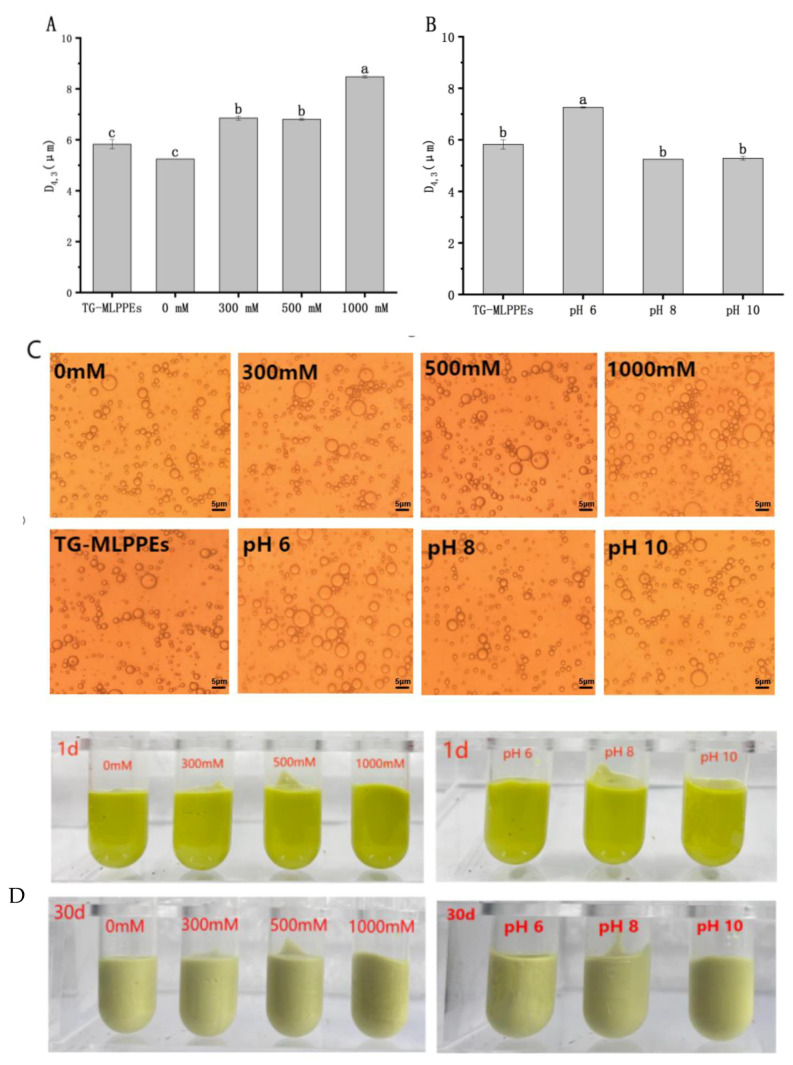
Droplet size (**A**,**B**), microscopic morphology under optical microscope (**C**), and emulsion appearance before and after 30 days of storage under different ionic strength and pH conditions (**D**). Different lowercase letters denote significant differences (*p* < 0.05).

**Figure 2 foods-13-03939-f002:**
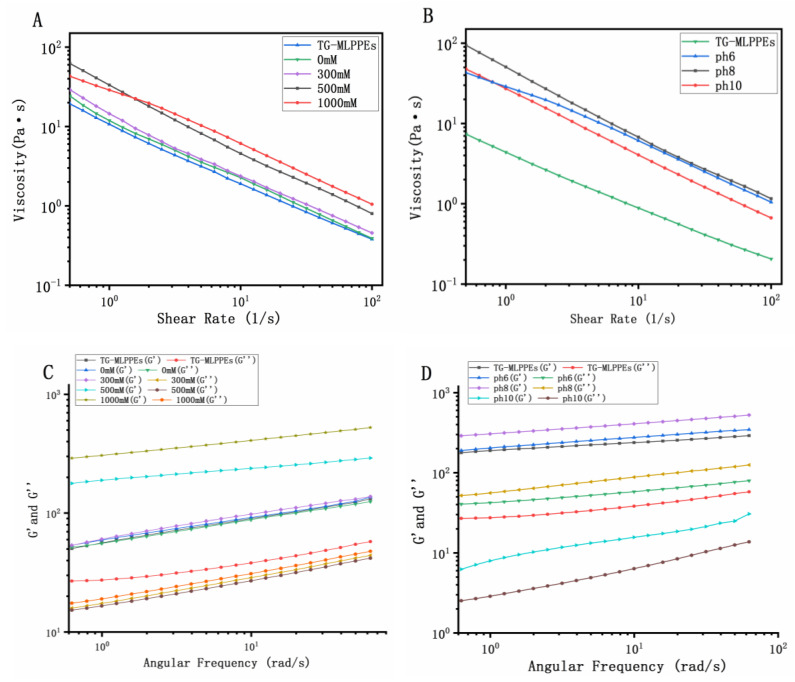
Effect of shear rate on apparent viscosity of emulsions (**A**,**B**). Effect of frequency on emulsions’ storage modulus (G′) and loss modulus (G′′) (**C**,**D**).

**Figure 3 foods-13-03939-f003:**
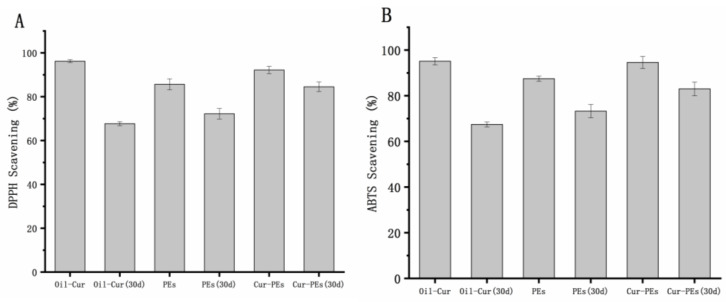
The effect of curcumin on the DPPH radical (**A**) and ABTS radical (**B**) scavenging activity of TG-MLPPs (PEs in the figure is the emulsion without curcumin).

**Figure 4 foods-13-03939-f004:**
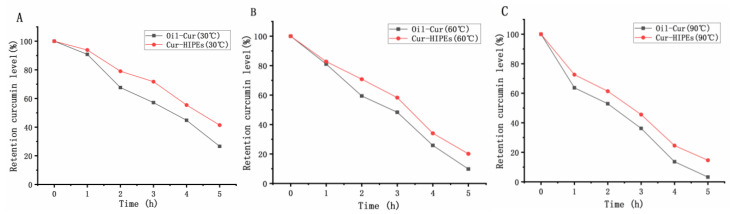
Thermal stability of high-internal-phase Pickering emulsions loaded with curcumin ((**A**) 30 °C; (**B**) 60 °C; (**C**) 90 °C).

**Figure 5 foods-13-03939-f005:**
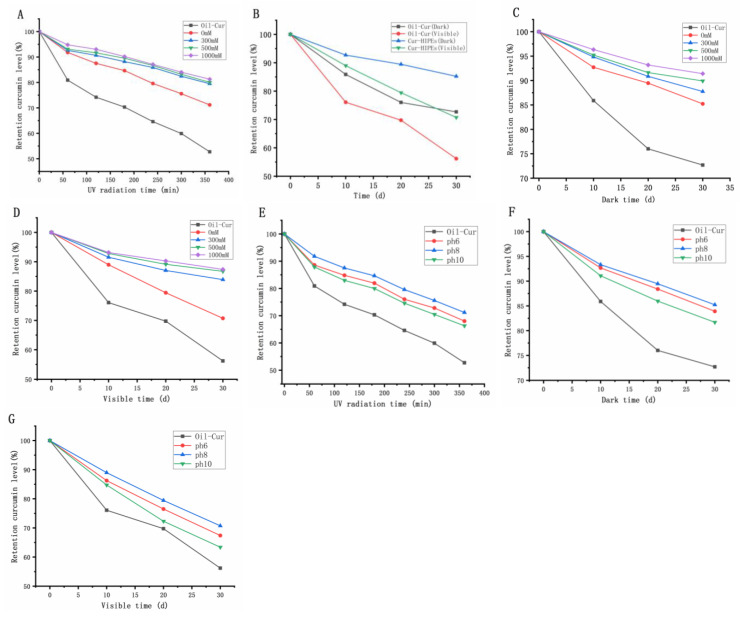
Effects of Pickering emulsion storage in UV (**A**), dark (**C**), and visible light (**D**) on curcumin retention under different ionic strength conditions; Effects of Pickering emulsion storage in UV (**E**), dark (**F**), and visible light (**G**) on curcumin retention under different pH conditions; Effects of Oil-cur and Cur-HIPEs storage in dark and visible light on curcumin retention (**B**).

**Figure 6 foods-13-03939-f006:**
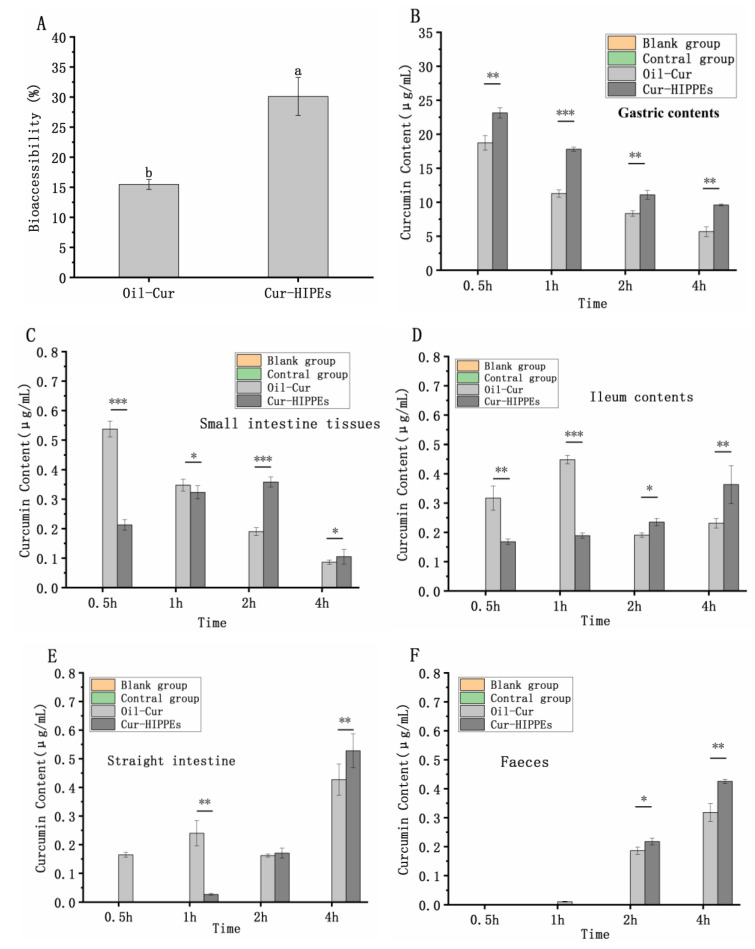
HPLC of bioaccessibility of curcumin (**A**). Curcumin content in stomach contents of mice (**B**). Curcumin content in small intestine of mice (**C**). Curcumin content in rectum of mice (**D**). Curcumin content in mouse ileum (**E**). Curcumin content in mouse feces (**F**). Different lowercase letters and * denotes statistically significant differences * *p* < 0.05; ** *p* < 0.01; *** *p* < 0.001.

**Table 1 foods-13-03939-t001:** First-order kinetic model parameters and half-life of TG-MLP-stabilized Pickering emulsion under UV, dark, and visible light storage induced by curcumin degradation under different ionic strengths.

Sample	K (min^−1^)			R^2^			t_1/2_/min		
	UV	Dark	Vis	UV	Dark	Vis	UV	Dark	Vis
Oil-Cur	1.72 × 10^−3 d^	1.25 × 10^−2 e^	1.82 × 10^−2 e^	0.994	0.993	0.966	40.3	55.4	38.1
0 mM	0.84 × 10^−3 c^	0.52 × 10^−2 d^	1.06 × 10^−2 d^	0.984	0.975	0.996	82.5	133.3	65.4
300 mM	0.58 × 10^−3 b^	0.44 × 10^−2 c^	0.58 × 10^−2 c^	0.971	0.991	0.960	119.5	157.5	119.5
500 mM	0.57 × 10^−3 b^	0.36 × 10^−2 b^	0.47 × 10^−2 b^	0.940	0.965	0.948	121.6	192.5	147.5
1000 mM	0.55 × 10^−3 a^	0.30 × 10^−2 a^	0.44 × 10^−2 a^	0.992	0.981	0.955	126.0	231.0	157.5

Note: Values are mean ± standard deviation (*n* = 3). Different superscript letters in same column indicated significant differences (*p* < 0.05).

**Table 2 foods-13-03939-t002:** First-order kinetic model parameters and half-life of TG-MLP-stabilized Pickering emulsion under UV, dark, and visible light storage induced by curcumin degradation under different pH conditions.

Sample	K (min^−1^)			R^2^			t_1/2_/min		
	UV	Dark	Vis	UV	Dark	Vis	UV	Dark	Vis
Oil-Cur	1.72 × 10^−3 d^	1.25 × 10^−2 d^	1.82 × 10^−2 d^	0.994	0.993	0.966	40.3	55.4	38.1
pH 6	0.97 × 10^−3 b^	0.58 × 10^−2 b^	1.30 × 10^−2 b^	0.976	0.987	0.998	71.4	119.5	53.3
pH 8	0.90 × 10^−3 a^	0.52 × 10^−2 a^	1.15 × 10^−2 a^	0.991	0.988	0.999	77.0	133.3	60.2
pH 10	1.06 × 10^−3 c^	0.67 × 10^−2 c^	1.53 × 10^−2 c^	0.976	0.978	0.997	65.4	103.4	45.3

Note: Values are mean ± standard deviation (*n* = 3). Different lowercase letters denote significant differences (*p* < 0.05).

## Data Availability

The original contributions presented in this study are included in the article/Appendix A. Further inquiries can be directed to the corresponding author.

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
