# Peer review of "Enhanced Bioaccessibility and Antioxidant Activity of Curcumin from Transglutaminase Cross-Linked Mulberry Leaf Protein-Stabilized High-Internal-Phase Pickering Emulsion: In Vivo and In Vitro Studies"

_foods, 2024, doi:10.3390/foods13233939_

Round 1

Reviewer 1 Report

Comments and Suggestions for Authors

Overall, the research article is well written and the main points are well addressed. I suggest though to add and focus more on how these findings can used and the field.

Abstract

The language can be simplified the make the reading more fluid. "serve as an excellent protective and efficient delivery system": "offer excellent protection and delivery."

Introduction:

Add a brief explanation of how high internal phase Pickering emulsions (HIPEs) differ from other emulsion types.

More details about the novelty of mulberry leaf protein (MLP) or why it's chosen over other protein sources could be highlighted early on.

I think you have confused the terms “bioavailability” and “bioaccesibbility” please check and correct through all the manuscript.

Methods:

Add more information about the preparation of TG-MLP and these conditions (e.g., pH, temperature) were selected.

Revise and correct in metric units.

Results and Discussion:

It should be better to discuss and compare your results with the literature and give more emphasis on the practical implications of these findings.

Can you describe why the enhanced antioxidant properties or improved bioaccessibility of curcumin imply for the food or pharmaceutical industry.

In some diagrams and figures the statistical differences are not present or needs more clarity.

Conclusion:

Please improve and expand the conclusion section with highlighting the future directions or applications of this work, or potential commercialization.

Comments on the Quality of English Language

The language can be simplified and you should add some transition sentences between sections to enhance the flow, particularly between the results and discussion sections. Although the title is concise, it is too long and can be more direct.  

Author Response

The language can be simplified the make the reading more fluid. "serve as an excellent protective and efficient delivery system": "offer excellent protection and delivery."

Thank you for your comment. We have reviewed the entire manuscript and refined the language to improve readability. Please see line 27、28、31 et al. (in red colour).

Add a brief explanation of how high internal phase Pickering emulsions (HIPEs) differ from other emulsion types.

Thank you for your comment. We have included a brief explanation on HIPEs. Please see line 37-40 (in red colour).

More details about the novelty of mulberry leaf protein (MLP) or why it's chosen over other protein sources could be highlighted early on.

Thank you for your comment. We have corrected according to your suggestion. Please see line 59-61 and 68-72 (in red colour).

I think you have confused the terms bioavailability and bioaccesibbility please check and correct through all the manuscript.

Thank you for your comment. We have corrected 'bioavailability' to 'bioaccessibility' throughout the manuscript.

Add more information about the preparation of TG-MLP and these conditions (e.g., pH, temperature) were selected.

Thank you for your comment. We have added detailed information about the preparation of TG-MLP. Please see line 163-166 (in red colour).

Revise and correct in metric units.

Thank you for your comment. We have corrected all units to the metric system in the manuscript.

It should be better to discuss and compare your results with the literature and give more emphasis on the practical implications of these findings.

Thank you for your comment. We have added a discussion and comparison with the existing literature. Please see line 548-551,591-592,650-651 (in red colour).

Can you describe why the enhanced antioxidant properties or improved bioaccessibility of curcumin imply for the food or pharmaceutical industry.

Thank you for your comment. We have included a description based on your suggestion. Please see line 437-440 and 97-100 (in red colour).

In some diagrams and figures the statistical differences are not present or needs more clarity.

Thank you for your comment. We have made changes to the figures. Please see Figure 3.

Please improve and expand the conclusion section with highlighting the future directions or applications of this work, or potential commercialization.

Thank you for your comment. We have included additional content regarding the future directions and potential applications of this work, as well as its possible commercialization. Please see line 669-679 (in red colour).

Reviewer 2 Report

Comments and Suggestions for Authors

This manuscript is well organized and presents good results. However, in order to improve its quality, some modifications are necessary, as indicated below:

- Conduct a study of the stability by heating and content of active substances using, for example, antioxidative activity or mass spectrometry;

- In Figure 1C, place the scale bar on the micrographs;

- The quality of the figures should be improved. In many graphs, the numbers and axis titles are too small and in low resolution, making proper reading impossible.

- Zeta potential values ​​should be made available in the article instead of in the supplementary material.

Author Response

- Conduct a study of the stability by heating and content of active substances using, for example, antioxidative activity or mass spectrometry;

We have indeed conducted a thermal stability analysis in our study. Please refer to Section 3.4, 'Thermal Stability,' for more details. Please see line 196-200 (in red colour).

- In Figure 1C, place the scale bar on the micrographs;

Thank you for your comment. We have updated the figure accordingly. Please see Figure 1C.

- The quality of the figures should be improved. In many graphs, the numbers and axis titles are too small and in low resolution, making proper reading impossible.

Thank you for your comment. We have updated the figure accordingly.

- Zeta potential values should be made available in the article instead of in the supplementary material.

Thank you for your comment. I modified the method. Please see line 195-199 (in red colour).
